# Metrics-Based Comparison of OWL and XML for Representing and Querying Cognitive Radio Capabilities

**Yanji Chen** [1], **Mieczyslaw M. Kokar** [2,3,*], **Jakub Moskal** [3] and **Kaushik R. Chowdhury** [2]

1. Google, Inc., 1600 Amphitheatre Pkwy, Mountain View, CA 94043, USA
2. Department of Electrical and Computer Engineering, Northeastern University, Boston, MA 02115, USA
3. VIStology, Inc., Framingham, MA 01701, USA
* Correspondence: mkokar@ece.neu.edu

**Abstract:** Collaborative spectrum access requires wireless devices to perform spectrum-related tasks (such as sensing) on request from other nodes. Thus, while joining the network, they need to inform neighboring devices and/or the central coordinator of their capabilities. During the operational phase, nodes may request other permissions from the the controller, like the opportunity to transmit according to the current policies and spectrum availability. To achieve such coordinated behavior, all associated devices within the network need a language for describing radio capabilities, requests, scenarios, policies, and spectrum availability. In this paper, we present a thorough comparison of the use of two candidate languages—Web Ontology Language (OWL) and eXtensible Markup Language (XML)—for such purposes. Towards this goal, we propose an evaluation method for automating quantitative comparisons with metrics such as precision, recall, device registration, and the query response time. The requests are expressed in both SPARQL Protocol and RDF Query Language (SPARQL) and XML Query Language (XQuery), whereas the device capabilities are expressed in both OWL and XML. The evaluation results demonstrate the advantages of using OWL semantics to improve the quality of matching results over XML. We also discuss how the evaluation method can be applicable to other scenarios where knowledge, datasets, and queries require richer expressiveness and semantics.

**Keywords:** OWL; XML; language comparison; evaluation method; cognitive radio; quantitative metrics

## 1. Introduction

### 1.1. Problem Scenario Descriptions

A wireless network of reconfigurable software-defined radios (SDRs) can be considered a distributed computing system, as these SDRs possess the ability to perform various sensing and computation tasks requested by applications running on other similar nodes. For this high level of coordination, SDRs (these intelligent SDRs are referred to as 'cognitive radios', or simply as 'radios' henceforth) need to perform various simple to complex functions. Thus, they need to determine how specific capabilities may best address the issued queries, e.g., individual radios may need to inform the network controller of their capabilities, and applications may issue requests for services and the controller then can match the radio capabilities against the requests. Following this, a subset of radios can be invoked to perform the requested services. Similarly, radios may request permission to transmit, and such requests then need to be matched against policies and available resources (e.g., availability of the spectrum). In all such scenarios, the requests need to be compared with the radio capabilities in order to decide which devices can satisfy a specific request.

Figure 1 shows a Unified Modeling Language (UML) diagram for some sample use-case scenarios. 'Device' and 'Application' are two actors who participate in the use cases of

the system running on the network. A device registers its capabilities to the network, as indicated by *Register Device* use case. After registering devices to the controller, whenever the latter receives a request by a device or an application for a service, it processes the request and then returns the results of the processing back to the requester. This functionality is captured by the *Request Service*, which includes the *Process Service* use case denoted by the dotted arrow from *Request Service* to *Process Service*. This arrow is annotated by the «include» label which, in UML terminology, implies that the use case at the tail end of the dotted arrow completely reuses all of the steps included for that case [1]. We can also observe from the figure that there are some special types of *Register Device* and *Request Service* use cases expressed by the inheritance relation, with the generalization arrow pointing to the more general use case from the more specific use case. This indicates that more than one language can be used to express device capabilities and application requests.

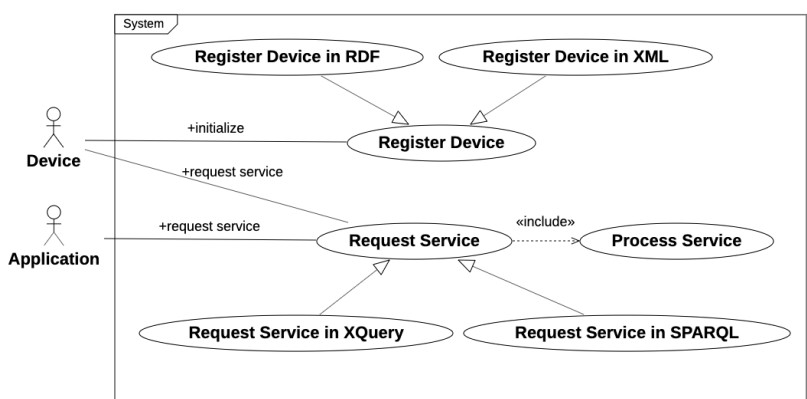

**Figure 1.** Problem scenarios—UML use case diagram.

Since both requests and device capabilities need to be formulated in a language that all devices understand, the main contribution of this paper is to identify a language that can describe the requests and radio capabilities. Such a language must be interpretable by devices, applications, and networks and it must have sufficient expressiveness to deal with various scenarios, requests, and device descriptions. The language must have precise semantics so that the meaning of the requests is precise and the devices can derive sound and complete results in response to the requests. Finally, the meaning of the returned results should be understandable by the requesters.

### 1.2. Approach

While we considered different candidate options, we advocate for the use of Web Ontology Language (OWL) [2] in this paper. OWL is a formal language (with formal syntax and formal semantics) designed to represent rich and complex knowledge about things, groups of things, and relations between things.

Though there are many advantages of using the semantics of the OWL language for representing and querying radio capabilities, it also comes with a price. For instance, there are time and space costs for deriving inferred facts with inference rules processed by an inference engine. When data need to be manipulated often, the inference engine should run every time a change is made to the data. We comprehensively explored the pros and cons of OWL versus eXtensible Markup Language (XML) [3] based schemas that match device descriptions against application requests in a quantitative way. While it is clear that OWL captures the semantics, the real question is how much the use of OWL semantics is advntageous for a specific application, versus an application that uses just XML. For this, we propose a metrics-based evaluation method to compare the two approaches. We performed experiments with the proposed method to prove the feasibility and correctness of the approach. To ensure that the evaluation results reflect practical concerns, our investigation was guided by the following principles:

1.  The comparison must be reasonably fair. Knowledge, datasets, and queries used to evaluate each of the two approaches must be comparable (equivalent).
2.  The samples of knowledge, data, and queries used in the experiment should provide good coverage of the space of knowledge, device descriptions, and query properties. Test samples should be highly diversified and representative of the whole tested space of devices and queries.
3.  The method must have access to the ground truth of each application request for the purpose of computing evaluation metrics.
4.  The same metrics should be used to evaluate the results of each of the two approaches.
5.  The evaluation must be scalable and extensible. The evaluation should not only be capable of dealing with different sizes of datasets and queries but should also be extensible for any updates with respect to the knowledge base, data, queries, and metrics.

### 1.3. Contributions and Paper Organization

The main contributions made in this paper include the following:

1.  We propose a method for evaluating the use of OWL-based and XML-based approaches to represent and query cognitive radio capabilities with quantitative metrics.
2.  We justify how the method satisfies all of the evaluation principles and why the techniques used in the method are a good fit to our problem.
3.  We analyze the evaluation results and provide recommendations for the selection of the best approach depending on the scenario.

The remainder of this paper is structured as follows: Section 2 summarizes related work, and Section 3 presents the reasons for choosing XML and OWL as the two candidate languages that we probe deeper into in the rest of the paper. Section 4 conceptualizes and formulates the problem. The evaluation method, results, and analysis are presented in detail in Section 5, followed by the justification of the method against satisfaction of the evaluation principles in Section 6. Section 7 concludes the paper.

## 2. Related Work

To the best of our knowledge, we are the first group to propose a comprehensive and quantitative assessment method for the comparison of the use of OWL and XML for modeling the Radio Frequency (RF) domain. The most closely related efforts address only portions of the entire problem, which we classified into the following categories: (1) automatic device description generation, (2) automatic device capabilities query generation, (3) data transformation between the Resource Description Framework (RDF) [4,5]/OWL and XML, and (4) query transformation between the SPARQL Protocol and RDF Query Language (SPARQL) [6] and XML Query Language (XQuery) [7]. Since the first two categories are fully addressed in [8,9], this section is focused entirely on the latter two.

### 2.1. Data Transformation between RDF/OWL and XML

The related work in this category, further classified into three groups, (1) XML to RDF/OWL, (2) RDF/OWL to XML, and (3) bidirectional transformation, is summarized in Table 1. The values in the second column indicate the types of support used for the schema transformation with the lack of such support being indicated by "N/A".

**Table 1.** Methods used for the transformation between RDF/OWL and XML.

| Related Work | Schema Transformation |
|---|---|
| (1) *XML → RDF/OWL* | |
| Ferdinand et al. [10] | XML Schema → OWL-DL |
| Garcia et al. [11] | XML Schema → OWL-FULL |
| Bohring et al. [12] | XML Schema → OWL-DL |
| JXML2OWL [13,14] | N/A |
| DTD2OWL [15] | DTD → OWL-DL |
| * Thuy et al. [16] | XML Schema → OWL-DL |
| Breitling [17] | N/A |
| * GRDDL [18] | N/A |
| * XMLtoRDF [19] | N/A |
| * Droop et al. [20,21] | N/A |
| XMLMaster [22] | N/A |
| EXCO [23] | XML Schema → OWL-DL |
| (2) *RDF/OWL → XML* | |
| XSLT + SPARQL [24] | N/A |
| (3) *Bidirectional* | |
| Gloze [25] | N/A |
| Miletic et al. [26] | XML Schema → OWL-DL |
| * SAWSDL [27] | N/A |
| * XSPARQL [28,29] | N/A |

\* Semi-automatic data transformation that requires user intervention.

In the first group (XML to RDF/OWL), the methods that fully automate the transformation typically rely on predefined, generic mapping rules that are encoded as XSLT stylesheets [30]. While the research presented in [10–12,15,16,23] relies on mappings developed at the schema level, in [17], mapping is defined at the data level. These transformations retain the original structure of the input XML, yielding RDF/OWL data that usually do not properly convey the intended semantics. Other methods in this group require either additional input or some form of user interaction. In [13,14], a graphical interface is relied on by the users to define the schema mapping rules, and this produces an XSLT for automatic data transformation. In [18], a mechanism for extracting RDF/OWL data from XML is defined. This involves declaring that XML data are compatible with RDF/OWL via linking to algorithms (typically XSLT). In [20,21], XML data are transformed to RDF by performing a depth-first search traversal of the XML tree. In addition to the XML data, in [19], ontology and a schema are required for ontology mapping by the user. In [22], user-provided mapping expressed in a domain-specific mapping language is employed.

For the second group (RDF/OWL to XML), in [24], an extension to the XSLT function set was proposed. This allows SPARQL queries to be embedded into XSLT which, in turn, provides a new platform for scripting and transforming RDF into XML.

For the third group (bidirectional), in [25], the transformation is provided based on the XML schema, and in [26], the RDF-based Semantic Mediation approach is used. In [27], mechanisms are provided for annotating XML schema elements (especially within WSDL definitions) that can be leveraged for automatic and bidirectional transformation. In [28,29], XSPARQL, a combination of XQuery and SPARQL with one language first extracting the native input data through queries and the other used for transformation to the target language through update statements, is defined.

The bidirectional methods are the most relevant to the work described in this paper, since they may better satisfy the fairness principle of the evaluation method (Section 1.2). However, in our work, instead of transforming between the two languages, we relied on the XML-based serialization of RDF/OWL, which can be treated as XML and RDF/OWL at the same time (Section 6).

### 2.2. Query Transformation between SPARQL and XQuery

In [31], a method is presented for providing SPARQL endpoints over XML data by integrating a schema into the ontology transformation component and automatically translating SPARQL queries to XQuery. This method is limited to queries that do not include OWL individuals, since the translation operates at the ontology/schema level only. In [32,33], a method for the complete and correct translation of SPARQL to XQuery isdescribed without any assumptions made about the schema or particular workload. Full implementation of the method (xql2xquery) is publicly available, and because of its good coverage of the SPARQL query features, it was adopted within the evaluation processes described in this paper.

### 3. Motivation for OWL vs. XML

There are many languages that could satisfy the requirements outlined in the previous section. This section discusses the motivation behind the selection of the two approaches evaluated in this paper.

First, we were interested in evaluating how the semantics affects the performance with respect to data modeling and the querying of cognitive radio capabilities. Since OWL has computer processable semantics (i.e., it belongs to a group of *formal languages*—languages with formal syntax and formal semantics) [34], the consistency of device descriptions represented in OWL can be checked first. Then, new facts can be derived from a given set of facts using the inference rules executed by an inference engine. The derived facts are guaranteed to be consistent with the knowledge base, i.e., its derivation process is sound. In contrast, other languages such as XML Schema Definition Language (XSD) [35,36]/XML or relation/tuple are either semi-structured or structured and do not have much of the semantic richness, require all facts about devices to be explicitly expressed, and cannot generate inferred facts without dedicated software. Therefore, the OWL-based approach was chosen as a baseline that plays a significant role in the matching process, especially in establishing the query ground truth for the evaluation of the quality of the matching results.

Second, we considered the XML-based approach rather than other approaches, such as the relational table approach, since the former better addresses the language requirements defined by the use cases described in Section 1.1. XML is more flexible than the relational table approach with respect to the updating of the structure of data. Whenever there is a need to update the structure of the data, it is much easier to make changes to the XML data than to a relational table, since the latter would require a change in the relational database schema, which would have to be followed by the restructuring of the whole database. Additionally, XML data are self-describing, while relational data are not [37]. An XML document contains not only the data, but also a tagging for the data that represent what it is [37]. With the relational model, the content of the data is defined by its column definition [37]. All data in a column must have the same type of data [37]. The flexibility feature is very critical in our problem, since the data representation language for our use cases should be flexible enough to deal with various data updates, especially for cases where devices with new types or capabilities are registered with the system. Therefore, in this paper, we only considered OWL-based and XML-based approaches as being appropriate to address the problem.

Each of these two approaches uses other languages for representing data and queries. The languages used in the two approaches are listed in Table 2. They are categorized into three levels [38]: (a) languages for modeling background knowledge (the schema level); (b) languages for representing device descriptions (the data level); and (c) languages for requests for services (the query level).

**Table 2.** Languages used in the two approaches for each level.

|  | OWL-Based | XML-Based |
| --- | --- | --- |
| **Background Knowledge (Schema Level)** | OWL | XSD |
| **Device Descriptions (Data Level)** | RDF/OWL | XML |
| **Requests for Services (Query Level)** | SPARQL | XQuery |

### 3.1. Web Ontology Language (OWL)

OWL descriptions can be viewed as consisting of two parts—descriptions of the concepts and then facts that are *instances* of these concepts. Concepts are analogous to database schema, while instances are analogous to tuples in database tables. The concepts are expressed in OWL, while the instances are expressed as RDF triples. OWL representations are called *ontologies*. The term ontology emerges from artificial intelligence and conveys the syntax and semantics of concepts and their relationships in a formal, declarative, and computer-understandable way [39]. An ontology for a radio communication domain serves as the base for representing device descriptions. Data are stored in, and retrieved as, triples in an RDF store, also called a *triple store*. Application requests are expressed in SPARQL. SPARQL is a set of specifications that provide languages and protocols to query and manipulate RDF graph content on the Web or in an RDF store [6]. Due to the fact that OWL has formal semantics, the data annotated in OWL can be processed by any *inference engine* (or *reasoner*) that conforms with the OWL semantics to derive the facts that are implicitly contained by the explicitly encoded facts. Thus querying is applied to the extended set of facts after the inference step.

### 3.2. eXtensible Markup Language (XML)

The second approach is to use the XML-based technology, i.e., use XSD as a base for expressing types and then use XML to describe instance XML data about the environment and the resources. There are two main approaches to the storage of XML documents—as an XML enabled database or as a native XML database. The former is a relational database that transfers data between XML documents and tuples, whereas the latter stores XML data directly [40]. Device descriptions can be represented in XML. They can be queried, transformed, exported, and returned to application requests expressed in XQuery—the "native" query language for XML. Application requests represented as XQuery queries are processed by the XQuery processor which matches XML data against the query and returns the matching results.

### 4. Problem Formulation

We conceptualize and formalize the problem in this section. Our objective was to derive metrics for the matching of the descriptions of device capabilities against requests for their services for the two approaches and provide recommendations on the usability of the two approaches in the RF domain based on these metrics.

Figure 2 is a data flow diagram that represents the evaluation process at a high level. As shown in this figure, a set of device descriptions (and their capabilities) is generated and represented in the respective data representation languages, i.e., XML in the XML-based approach and RDF/OWL in the OWL-based approach. The descriptions of device capabilities are stored in respective datastores. Requests for device capabilities are generated and expressed in two query languages—XQuery in the XML-based approach and SPARQL in the OWL-based approach. The requests represented in the respective query languages are forwarded to the appropriate matchers for matching of the device capabilities against the requests. The matching results of each query expression returned from the matchers are then evaluated by comparing with the ground truth of the matching devices against the query, and metrics are computed.

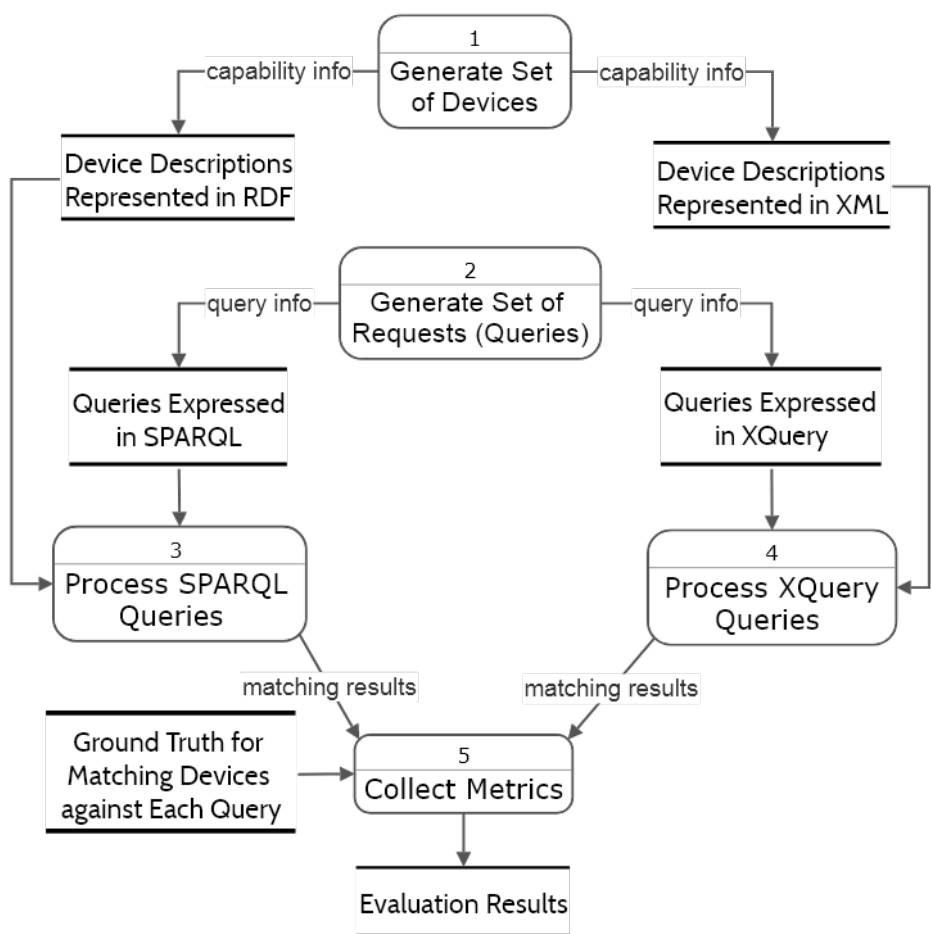

**Figure 2.** Visual representation of the evaluation process.

*4.1. Problem Formalization*

4.1.1. Notations

For the purpose of formalizing the problem, the following notations are introduced.

- $D = \{D_1, D_2, \ldots, D_n\}$—set of $n$ devices
- $D^{xml} = \{D_1^{xml}, D_2^{xml}, \ldots, D_n^{xml}\}$—set of $n$ descriptions of device capabilities represented in XML
- $B = \{B_0, B_1, \ldots, B_k\}$-set of $k+1$ knowledge representations of $D$ in OWL, where $B_0 \in B$ is the most complete representation and $B_i \in \{B_1, \ldots, B_k\}, 1 \leq i \leq k$ are progressively less complete representations s.t. each $B_i$ does not include at least one type of OWL 2 RL axiom with respect to $B_{i-1}$
- $D^{owl} = \{D_1^{owl}, D_2^{owl}, \ldots, D_n^{owl}\}$—set of $n$ descriptions of device capabilities from $D$ in RDF/OWL
- $Q = \{Q_1, Q_2, \ldots, Q_m\}$—set of $m$ requests (queries) against device capabilities from $D$
- $Q^{xml} = \{Q_1^{xml}, Q_2^{xml}, \ldots, Q_m^{xml}\}$—set of $m$ requests expressed in XQuery
- $Q^{owl} = \{Q_1^{owl}, Q_2^{owl}, \ldots, Q_m^{owl}\}$—set of $m$ requests expressed in SPARQL
- $f^{xml}$—function that takes a set of device descriptions and a query in XML and returns a set of devices that match the query:

$$f^{xml} : (2^{D^{xml}}, Q^{xml}) \rightarrow 2^D \qquad (1)$$

- $f^{owl}$—function that takes a set of device descriptions, a background knowledge representation and a query in OWL and returns a set of devices that match the query):

$$f^{owl} : (2^{D^{owl}}, B, Q^{owl}) \rightarrow 2^D \tag{2}$$

- $g^{xml}$—ground truth function that takes a set of device descriptions and a query in XML and returns a set of devices that the query is expected to return:

$$g^{xml} : (2^{D^{xml}}, Q^{xml}) \rightarrow 2^D \tag{3}$$

- $g^{owl}$—function that takes a set of device descriptions and a query in OWL and returns a set of devices that the query is expected to return:

$$g^{owl} : (2^{D^{owl}}, Q^{owl}) \rightarrow 2^D \tag{4}$$

- $m^{xml}$—function that compares two functions, a query function and a ground truth function in XML, and returns the value of the metric from [0, 1]:

$$m^{xml} : (f^{xml}(2^{D^{xml}}, Q^{xml}), g^{xml}(2^{D^{xml}}, Q^{xml})) \rightarrow [0, 1] \tag{5}$$

- $m^{owl}$—function that compares two functions, a query function and a ground truth function in OWL, and returns the value of the metric from [0, 1]:

$$m^{owl} : (f^{owl}(2^{D^{owl}}, B, Q^{owl}), g^{owl}(2^{D^{owl}}, Q^{owl})) \rightarrow [0, 1] \tag{6}$$

The metrics $m^{xml}$ and $m^{owl}$ introduced above are just patterns used for defining concrete metrics. The concrete metrics formalization and definitions are presented in Section 4.2.

### 4.1.2. Examples

To illustrate the relations between devices/queries and their representations in the two approaches with the presented notations, we provide two examples. One is a device with its configuration and its representations in the two approaches. The other is a query example in natural language as well as its representations in the respective query languages of the two approaches.

**A Device Description Example**: An example of the device $D_i \in D$ named *device_instance* is described in Table 3. The rows show the parameters of this device.

**Table 3.** A device example.

| | Specifications | Parameters |
|---|---|---|
| Device Name | | device_instance |
| Series | | USRP_N200 |
| | Bandwidth | 200 kHz |
| Spectrum Sensing | Frequency Range | 80–120 MHz |
| | Sensing Time | 1 s |
| Transmitting | Max Output Power | 0.01 W |

Listing 1 ($D_i^{xml} \in D^{xml}$) and Listing 2 ($D_i^{owl} \in D^{owl}$) show this device represented in XML and OWL, respectively. In this example, the device name "device_instance" is mapped as an XML attribute in XML, whereas an OWL individual is mapped in OWL.

**Listing 1.** Device description example in XML.

```
<SupportsSpectrumSensing>
<Bandwidth units="kHz">200<Bandwidth>
<FrequencyRange>
<Min units="MHz">80</Min>
<Max units="MHz">120</Max>
</FrequencyRange>
<SensingTime units="Second">1.0</SensingTime>
</SupportsSpectrumSensing>
<SupportsTransmitting>
<MaxOutputPower units="Watt">0.01</MaxOutputPower>
</SupportsTransmitting>
</USRP_N200>
```

**A Query Representation Example**: Consider the following query expressed in natural language: *"Show all RF devices of the USRP network series whose maximum output power is no more than 1.0 Watt."* Listing 3 ($Q_i^{xml} \in Q^{xml}$) and Listing 4 ($Q_i^{owl} \in Q^{owl}$) represent this query in XQuery and SPARQL, respectively.

**Listing 2.** OWL device description example in Turtle syntax.

```
device_instance a USRP_N200;
hasCapability spectrumSensing, transmitting.
spectrumSensing a SpectrumSensing;
hasBandwidth [hasValue [hasDataValue 200; hasUoM kHz ]];
hasFrequencyRange [
hasMin [hasValue [hasDataValue 80; hasUoM MHz]];
hasMax [hasValue [hasDataValue 120; hasUoM MHz]]];
hasSensingTime [hasValue [hasDataValue 1.0; hasUoM Second]].
transmitting hasMaxOutputPower [
hasValue [hasDataValue 0.01; hasUoM Watt]].
```

**Listing 3.** An example of request representation in XQuery.

```
for $device in doc("devices.xml")/USRP_Nxxx
where $device/SupportsTransmitting/@units = "Watt"
and $device/SupportsTransmitting[MaxOutputPower<=1.0]
return data($device/@name)
```

**An OWL Inference Example**: The device example presented in Listings 1 and 2 does not match the query example shown in Listings 3 and 4. The issue here is that the device is described as USRP_N200, while the query is asking for USRP_Nxxx. While a human might know that USRP_N200 is a subtype of USRP_Nxxx, a query processor would not be able to match this request with the USRP_N200 device. However, if background knowledge $B_i \in B$ includes triple (*USRP_N200 rdfs:subClassOf USRP_Nxxx*), an inference engine that supports OWL 2 RL will be able to infer that (*device_instance a USRP_Nxxx*). The net result is then that a query engine that has support from an OWL reasoner will return a match. While this is a very simple example of the usability of OWL reasoning in the process of matching descriptions versus queries, the ontology may have many axioms about the specific types of devices, and thus the number of inferences can be quite large.

### 4.2. Metrics Formulation

The evaluation metrics whose signatures were defined in Section 4.1.1 are divided into two groups—query result quality metrics and query process performance metrics. The former includes the query result completeness, query result soundness, and a combined metric of the two, named F-Measure [41]. The latter contains the query response time and the load time metrics. Since all of the metrics require knowledge of the ground truth, the query ground truth function is introduced first. Since our objective was to assess the value

of automatic inference provided by OWL, we used the results returned by an OWL inference engine for the strongest ontology that we had, $B_0$, as the ground truth. This approach is justified by the fact that OWL is a formal language and because OWL reasoners are sound, i.e., they return only the true results, which will be as good as the ontology. Equation (7) shows how the ground truth is defined in terms of $f^{owl}(D^{owl}, B_0, Q_i^{owl})$ implemented by the OWL-based matcher described in Section 5.2.

$$
\begin{aligned}
g^{xml}(D^{xml}, Q_i^{xml}) &= g^{owl}(D^{owl}, Q_i^{owl}) \\
&= f^{owl}(D^{owl}, B_0, Q_i^{owl})
\end{aligned}
\tag{7}
$$

**Listing 4.** An example of a request representation in SPARQL.

```
SELECT DISTINCT ?device
WHERE {
?device a USRP_Nxxx; hasCapability [hasMaxOutputPower [
hasValue [hasDataValue ?dataValue; hasUoM Watt]]]
FILTER (?dataValue <= "1.0"^^xsd:double)
}
```

### 4.2.1. Quality Metrics

The *query result completeness* metric, referred to as the *recall metric* in information retrieval, measures the degree of how many of the expected results are returned by a query. The XML-based metric, $m_c^{xml}$, and the OWL-based metric, $m_c^{owl}$, are defined in (8) and (9), respectively.

The *query result soundness* metric measures the degree of how many query results returned are as expected out of all the returned results. In information retrieval, this metric is called the *precision* metric. The XML-based metric, $m_s^{xml}$, and the OWL-based metric, $m_s^{owl}$, are defined in (10) and (11), respectively.

The *F-Measure* [42] is a combined metric that measures the trade-off between the completeness and the soundness of the query results. The XML-based and OWL-based definitions of this metric are shown in (12) and (13), respectively. In (12), $m_c^{xml}()$ is short for $m_c^{xml}(f^{xml}(D^{xml}, Q_i^{xml}), g^{xml}(D^{xml}, Q_i^{xml}))$ introduced in (8), $m_s^{xml}()$ is short for $m_s^{xml}(f^{xml}(D^{xml}, Q_i^{xml}), g^{xml}(D^{xml}, Q_i^{xml}))$ introduced in (10). In (13), $m_c^{owl}()$ is short for $m_c^{owl}(f^{owl}(D^{owl}, B_j, Q_i^{owl}), g^{owl}(D^{owl}, Q_i^{owl}))$ introduced in (9), $m_s^{owl}()$ is short for $m_s^{owl}(f^{owl}(D^{owl}, B_j, Q_i^{owl}), g^{owl}(D^{owl}, Q_i^{owl}))$ introduced in (11). In both equations, $\beta \in [0, \infty]$ is a parameter that controls the balance between *precision* and *recall* [42]. In this paper, $\beta$ was set to 1 since we equally weighed the importance levels of *precision* and *recall*.

$$
\begin{aligned}
m_c^{xml}(f^{xml}(D^{xml}, Q_i^{xml}), \\
g^{xml}(D^{xml}, Q_i^{xml})) =
\end{aligned}
\begin{cases}
\dfrac{|f^{xml}(D^{xml}, Q_i^{xml}) \cap g^{xml}(D^{xml}, Q_i^{xml})|}{|g^{xml}(D^{xml}, Q_i^{xml})|}, & \text{if } g^{xml}(D^{xml}, Q_i^{xml}) \neq \varnothing \\
1, & \text{if } g^{xml}(D^{xml}, Q_i^{xml}) = f^{xml}(D^{xml}, Q_i^{xml}) = \varnothing \\
0, & \text{Others}
\end{cases}
\tag{8}
$$

$$
\begin{aligned}
m_c^{owl}(f^{owl}(D^{owl}, B_j, Q_i^{owl}), \\
g^{owl}(D^{owl}, Q_i^{owl})) =
\end{aligned}
\begin{cases}
\dfrac{|f^{owl}(D^{owl}, B_j, Q_i^{owl}) \cap g^{owl}(D^{owl}, Q_i^{owl})|}{|g^{owl}(D^{owl}, Q_i^{owl})|}, & \text{if } g^{owl}(D^{owl}, Q_i^{owl}) \neq \varnothing \\
1, & \text{if } g^{owl}(D^{owl}, Q_i^{owl}) = f^{owl}(D^{owl}, B_j, Q_i^{owl}) = \varnothing \\
0, & \text{Others}
\end{cases}
\tag{9}
$$

$$
\begin{aligned}
m_s^{xml}(f^{xml}(D^{xml}, Q_i^{xml}), \\
g^{xml}(D^{xml}, Q_i^{xml})) =
\end{aligned}
\begin{cases}
\dfrac{|f^{xml}(D^{xml}, Q_i^{xml}) \cap g^{xml}(D^{xml}, Q_i^{xml})|}{|f^{xml}(D^{xml}, Q_i^{xml})|}, & \text{if } f^{xml}(D^{xml}, Q_i^{xml}) \neq \varnothing \\
1, & \text{if } f^{xml}(D^{xml}, Q_i^{xml}) = g^{xml}(D^{xml}, Q_i^{xml}) = \varnothing \\
0, & \text{Others}
\end{cases}
\tag{10}
$$

$$m_s^{owl}(f^{owl}(D^{owl}, B_j, Q_i^{owl}), \\ g^{owl}(D^{owl}, Q_i^{owl})) = \begin{cases} \dfrac{|f^{owl}(D^{owl}, B_j, Q_i^{owl}) \cap g^{owl}(D^{owl}, Q_i^{owl})|}{|f^{owl}(D^{owl}, B_j, Q_i^{owl})|}, \text{if } f^{owl}(D^{owl}, B_j, Q_i^{owl}) \neq \varnothing \\ 1, \text{if } f^{owl}(D^{owl}, B_j, Q_i^{owl}) = g^{owl}(D^{owl}, Q_i^{owl}) = \varnothing \\ 0, \text{Others} \end{cases}$$

(11)

$$m_f^{xml}(m_c^{xml}(), m_s^{xml}()) = \begin{cases} \dfrac{(\beta^2 + 1) \cdot m_c^{xml}() \cdot m_s^{xml}()}{\beta^2 \cdot m_c^{xml}() + m_s^{xml}()}, \text{ if } m_c^{xml}() \neq 0 \vee m_f^{xml}() \neq 0 \\ 0, \text{ if } m_c^{xml}() = m_f^{xml}() = 0 \end{cases}$$

(12)

$$m_f^{owl}(m_c^{owl}(), m_s^{owl}()) = \begin{cases} \dfrac{(\beta^2 + 1) \cdot m_c^{owl}() \cdot m_s^{owl}()}{\beta^2 \cdot m_c^{owl}() + m_s^{owl}()}, \text{ if } m_c^{owl}() \neq 0 \vee m_f^{owl}() \neq 0 \\ 0, \text{ if } m_c^{owl}() = m_f^{owl}() = 0 \end{cases}$$

(13)

#### 4.2.2. Performance Metrics

The *device registration time* is the time taken for the registration of specified devices with repositories. Note that, in the OWL-based approach, besides the time taken for storing descriptions of the capabilities of the devices, the device registration time also includes the time for reasoning against both the device descriptions and the background knowledge for deriving inferred facts. There are two kinds of characteristics that will be collected using this metric. The first is the device registration time for the two approaches with specified test background knowledge representations $B_i \in B, 1 \leq i \leq k$ versus an increasing number of devices. The second is the device registration time for a specified number of devices in the OWL-based approach versus the test background knowledge representations $\{B_1, B_2, \ldots, B_k\}$. The objective of collecting these metrics is to compare the device registration times in the two approaches against the increasing number of devices and to show how the registration time of the specified devices varies with as the completeness of the background knowledge increases in the OWL-based approach.

For each query, the *query response time* is the processing time of the query required by the query processor of the system. We compared the average query response times of a specified number of queries in the two approaches with a specified test background knowledge representation $B_i \in B, 1 \leq i \leq k$ versus an increasing number of devices. Additionally, we compared the average query response time of a specified number of queries against a specified number of devices versus the test background knowledge representations $\{B_1, B_2, \ldots, B_k\}$ in the OWL-based approach.

### 5. Evaluation

In order to provide a comprehensive evaluation of the two approaches, a large pool of descriptions of radio capabilities and a wide variety of request types (queries) against the device descriptions in the languages of the two approaches are needed. Theoretically, such an investigation could be carried out experimentally using real radios and real queries or at least using data and queries collected from real radio networks. Unfortunately, these kinds of data and queries are not readily available for various reasons, primarily due to the privacy and security concerns. Thus, the only practical solution is to use synthetic data and queries instead. As a consequence, the method utilizes synthetic device descriptions and query expressions generated (randomly) by the components (RODG [8] and SQG [9]) that we developed.

#### 5.1. Evaluation Process

A high-level representation of the evaluation process is shown in Figure 2. This section describes the evaluation process in more detail. Figure 3 shows the concrete data flow in the evaluation process with the notations introduced in Section 4.1. Knowledge about a set of devices $D$ is modeled as the most complete ontology $B_0$. The method first automatically

generates device descriptions in RDF/XML [43] from $B_0$ using RODG. Since RDF/XML constitutes a base for OWL, while at the same time its syntax is XML, the device descriptions in RDF/XML can serve as data representations of $D$ for both the OWL-and XML-based approaches denoted by $D^{owl}$ and $D^{xml}$, respectively. Next, the method generates query expressions, $Q^{owl}$ and $Q^{xml}$, and requests about device capabilities for the two approaches. This is achieved through two procedures. First, the method automatically generates a set of random SPARQL queries as $Q^{owl}$ from $B_0$ and $D^{owl}$ using SQG. Then, the set of XQuery queries $Q^{xml}$ is obtained by converting a SPARQL query $Q^{owl}$ into an equivalent XQuery query in $Q^{xml}$ using the open source component named xql2xquery [32,33].

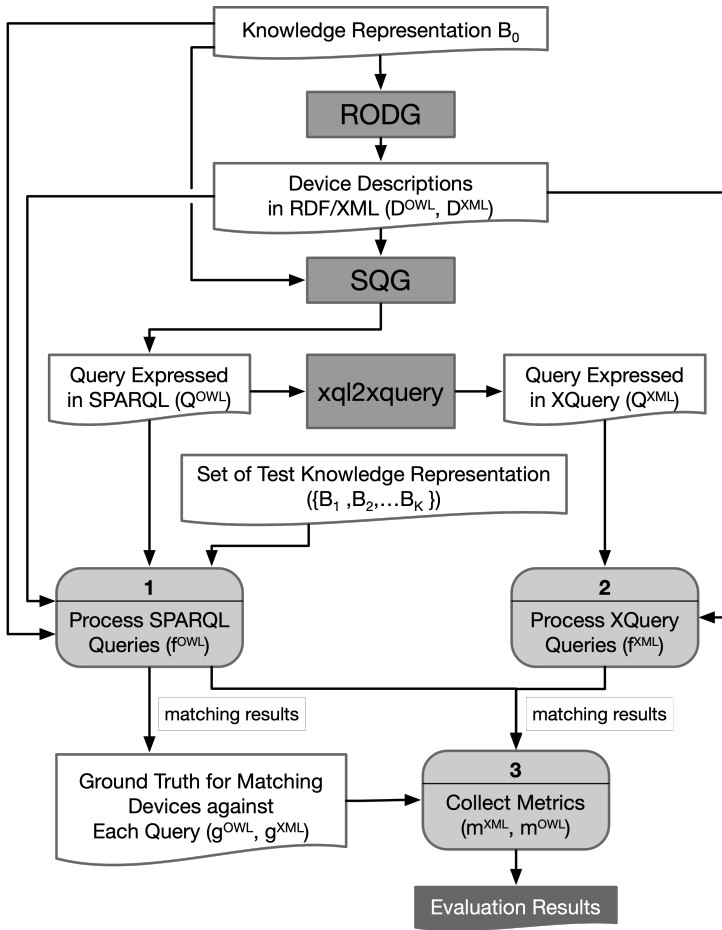

**Figure 3.** Data flow diagram of the evaluation process with the notations.

The queries are then submitted to the "matchers". The method uses DeVISor as the OWL-based matcher and BaseX as the XML-based matcher. The matchers include device registration. Note that, in the OWL-based approach, apart from the device descriptions, a background knowledge test ($B_i \in B$, $1 \leq i \leq k$) needs to be loaded into the OWL-based matcher (DeVISor) to derive inferred facts. Then, the processing of each query is executed for both approaches by the implemented query functions $f^{owl}$ and $f^{xml}$.

The matching results of each query expression returned from the matchers are evaluated by comparing them with the ground truth of the matching devices against the query. This is achieved by implementing all metric functions of the two approaches ($m^{xml}$ and $m^{owl}$). The implementation of the metric functions is shown in Algorithms 1–3. Additionally, the method collects metrics on the performance during the matching process. Finally, the metric values are derived and stored in an external file for further analysis.

---

**Algorithm 1:** CALC-RECALL($D_m$, $D_g$).

---

**Input:** $D_m$: matching devices of a query; $D_g$: Ground truth of the query
**Output:** $m_c$: recall metric value $\in [0, 1]$

1 **if** $|D_g| \neq 0$ **then**
2     $count \leftarrow 0$
3     **foreach** $D_{i,m} \in D_m$ **do**
4        **if** $D_{i,m} \in D_g$ **then**
5           $count \leftarrow count + 1$
6     $m_c \leftarrow \frac{count}{|D_g|}$
7 **else if** $|D_m| = 0$ **then**
8     $m_c \leftarrow 1$
9 **else**
10     $m_c \leftarrow 0$

---

**Algorithm 2:** CALC-PRECISION($D_m$, $D_g$).

---

**Input:** $D_m$: matching devices of a query; $D_g$: Ground truth of the query
**Output:** $m_s$: precision metric value $\in [0, 1]$

1 **if** $|D_m| \neq 0$ **then**
2     $count \leftarrow 0$
3     **foreach** $D_{i,m} \in D_m$ **do**
4        **if** $D_{i,m} \in D_g$ **then**
5           $count \leftarrow count + 1$
6     $m_s \leftarrow \frac{count}{|D_m|}$
7 **else if** $|D_g| = 0$ **then**
8     $m_s \leftarrow 1$
9 **else**
10     $m_s \leftarrow 0$

---

**Algorithm 3:** CALC-F-MEASURE($m_s$, $m_c$).

---

**Input:** $m_s$: precision metric value $\in [0, 1]$; $m_c$: recall metric value $\in [0, 1]$
**Output:** $m_f$: F-measure metric value $\in [0, 1]$

1 **if** $m_s \neq 0$ **or** $m_c \neq 0$ **then**
2     $m_f \leftarrow \frac{2 \times m_s \times m_c}{m_s + m_c}$
3 **else**
4     $m_f \leftarrow 0$

---

Some more details about the implementation of the evaluation process are presented in the next section.

### 5.2. OWL-Based Matcher—DeVISor

DeVISor [44] is a semantic matcher used in the OWL-based approach that is able to run on the network as a service and infer which devices are capable of satisfying a given request based on knowledge of the devices. Figure 4 presents the architecture of DeVISor. It is implemented as a web service and extended to support SPARQL. The design follows client–server architecture. On the client side, the DeVISor client provides an Application Programming Interface (API) to interact with application requests, and device descriptions needed to be registered. On the server side (shown in the rectangle with dashed line), it controls the process of the device (de)registration and matching device capabilities against the application requests. The support for data storage is provided by

Apache Jena Fuseki [45]. However, since the triple store is not equipped with the OWL inference, the DeVISor server uses BaseVISor [46], a forward-chaining inference engine specialized to handle facts in the form of RDF triples with support for OWL 2 RL [47] and XML Schema Datatypes, to make automatic inferences about any changes to the triple store. The following illustrates how DeVISor gets involved in the matching process with the associated implementations.

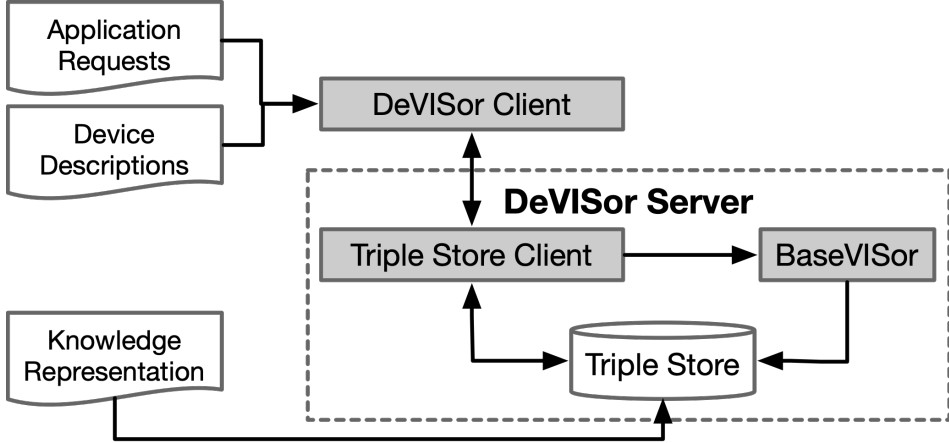

**Figure 4.** DeVISor architecture.

The first step taken during the matching process is to register devices. DeVISor supports such a feature by implementing an associate function (*runSparqlUpdateFromString(updateScript:String):void*), a function of the DeVISor client API which takes a SPARQL Update script as an input argument. Whenever there are RDF device descriptions needed to be registered, DeVISor wraps the dataset as a SPARQL Update script and invokes this function. After the script has been executed by the triple store, the dataset is inserted into the triple store as a set of triples. After that, the triple store returns all of the facts (including background knowledge) to the triple store client. BaseVISor loads the facts and executes the inference engine to derive inferred facts. The inferred facts are then inserted into the triple store.

After the devices have been registered, the next step is to send the application requests to DeVISor for matching. This feature is implemented by *runSparqlQueryFromFile(queryPath:String):String*, a function of the DeVISor client API, which is able to process a request as a SPARQL query script and return the query results in JSON format. When the function is invoked, DeVISor forwards the script to the triple store, and the triple store executes the script with the function and returns the results back. Note that BaseVISor is not involved in this process, since no inference is needed, and all of the inferred facts are already stored in the triple store. Since the results are in JSON format, the last step is to extract the matching devices from the results. The devices are identified by the Internationalized Resource Identifier (IRI) and are extracted by the binding value of the device variable in variable name/value pairs of the results.

*5.3. XML-Based Matcher—BaseX*

BaseX [48] is a robust, high-performance XML database engine and a fully compliant XQuery 3.1 processor with full support from the W3C Update and text extensions. It was selected as the XML-based matcher for matching devices represented in XML against application requests for services expressed in XQuery. Similar to DeVISor, BaseX follows the client–server architecture and also provides a client API [49] that supports multiple programming languages. The procedure of each step of the operation of BaseX is described below.

Device registration is implemented by calling a function of the BaseX client API, *add(path:String, input:InputStream):void*, which adds an XML document of the specified

device description into the database. Once the devices are registered, the next step is to send the application requests to BaseX for matching. Several functions are supported by the BaseX client API for XQuery query processing. The first step is to parse a XQuery query script and pass it as a parameter into the function *execute(command:String):String*. When the function is invoked, BaseX processes the query and returns the query results. The results are in XML format, and each device is identified by its IRI, so the method navigates through the nested structure of the results and extracts the values within the element tag named "xqllib:var".

*5.4. Computation of Metrics*

The query ground truth function was introduced in (7). According to this equation, given two representations, $D^{xml}$ and $D^{owl}$, of a set of devices $D$ and two query expressions, $Q_i^{xml}$ and $Q_i^{owl}$, of a request for matching device capabilities $Q_i \in Q$, the ground truth is assessed by taking the most complete ontology $B_0$ as a reference and using it for evaluating both approaches.

While this approach is not perfect, i.e., it would be better to have an independent source of ground truth, the simple fact is that such a source does not exist. While it would be possible to develop the ground truth by hand, such an approach is only applicable to cases where there is a relatively small set of device capabilities and a small set of queries. Since our intent was to assess the quality of matching on really large sets of device descriptions and queries, we had to come up with a solution that could be executed automatically without much participation by the human expert in the loop. While $B_0$ is advantageous for the OWL-based approach, the idea of using a descending chain of less and less complete ontologies makes this method relatively fair for both OWL- and XML-based approaches. Further discussions on fairness of the method are presented in Section 6.

The implementation of the query result completeness ((8) and (9)), soundness ((10) and (11)) and the F-measure ((12) and (13)) metrics are shown in Algorithms 1–3, respectively. It is worth noting that although XML-based and OWL-based approaches utilize their own notations to define the metrics, the definitions of the metrics for the two approaches are the same. Consequently, we used the same implementations for computing these metrics.

*5.5. Experiments*

We designed and implemented a proof-of-concept system to show the feasibility and correctness of the evaluation method (The source code is available at https://github.com/YankeeChen/evaluator, accessed on 15 November 2022). In this section, we present the experimental results obtained with the system.

5.5.1. Experimental Setup

All experiments were run on the MacBook Pro 2016 computer with the following parameters: Processor, 2.6 GHz quad-core Intel Core i7, Turbo Boost up to 3.5 GHz, with 6 MB shared L3 cache; Memory, 256 GB PCIe-based onboard SSD; and Storage, 16 GB of 2133 MHz LPDDR3 onboard memory.

In the experiments, the SDR ontology (https://github.com/YankeeChen/evaluator/blob/master/ontologies/SDROntology/BenchmarkOntology/SDR.owl, accessed on 15 November 2022) was selected as the background knowledge of the RF devices and set as the most complete knowledge representation, $B_0$. In order to assess the effect of the richness of the ontology on the quality of the matching process and performance in the OWL-based approach, we developed five progressively less complete cases of background knowledge, $\{B_1, B_2, B_3, B_4, B_5\}$, as the test ontologies (https://github.com/YankeeChen/evaluator/tree/master/ontologies/\SDROntology/TestOntology, accessed on 15 November 2022). The smaller (less complete) ontologies were created by removing some of the defining axioms from the previous ontology. Table 4 shows the axiom coverage of the ontologies. For simplicity, only the axioms that $B_0$ includes are listed. $B_5$ is not included in this table since it did not cover any of the axioms. More axioms defining the radio capabilities add

more constraints, and thus, the OWL reasoner is expected to be more precise when selecting devices that satisfy queries. The less complete definition implies that OWL reasoning can result in false negative classifications. This aspect of completeness is not directly correlated with the results of the XML-based approach, since it does not make use of logical inference.

**Table 4.** Axiom coverage of the ontologies.

| Axiom Type | $B_0$ | $B_1$ | $B_2$ | $B_3$ | $B_4$ |
|:---:|:---:|:---:|:---:|:---:|:---:|
| SubClassOf | ✓ | ✓ | ✓ | | |
| EquivalentClasses | ✓ | ✓ | ✓ | ✓ | ✓ |
| DisjointClasses | ✓ | ✓ | ✓ | ✓ | ✓ |
| DisjointUnion | ✓ | ✓ | ✓ | ✓ | ✓ |
| SubObjectPropertyOf | ✓ | ✓ | | | |
| EquivalentObjectProperties | ✓ | ✓ | ✓ | ✓ | ✓ |
| DisjointObjectProperties | ✓ | ✓ | ✓ | ✓ | ✓ |
| InverseObjectProperties | ✓ | ✓ | ✓ | ✓ | ✓ |
| ObjectPropertyDomain | ✓ | ✓ | ✓ | ✓ | |
| ObjectPropertyRange | ✓ | ✓ | ✓ | ✓ | |
| FunctionalObjectProperty | ✓ | ✓ | ✓ | ✓ | ✓ |
| SymmetricObjectProperty | ✓ | ✓ | ✓ | ✓ | ✓ |
| TransitiveObjectProperty | ✓ | ✓ | ✓ | ✓ | ✓ |
| SubDataPropertyOf | ✓ | | | | |
| EquivalentDataProperties | ✓ | ✓ | ✓ | ✓ | ✓ |
| DataPropertyDomain | ✓ | ✓ | ✓ | ✓ | ✓ |
| DataPropertyRange | ✓ | ✓ | ✓ | ✓ | ✓ |
| FunctionalDataProperty | ✓ | ✓ | ✓ | ✓ | ✓ |
| ClassAssertion | ✓ | ✓ | ✓ | ✓ | ✓ |
| ObjectPropertyAssertion | ✓ | ✓ | ✓ | ✓ | ✓ |
| DataPropertyAssertion | ✓ | ✓ | ✓ | ✓ | ✓ |

In the experiments, 1000, 2000, 3000, and 4000 device descriptions in RDF/XML generated by RODG [8] with the most complete knowledge base $B_0$ were selected as the test datasets for the two approaches. Four batches of 5000 queries in SPARQL and XQuery were selected as the test query sets for matching device capabilities against the test datasets. For each batch of queries, SPARQL queries were generated by SQG [9] with the most complete ontology $B_0$ and respective test datasets. The corresponding XQuery queries were generated by converting the SPARQL queries with xql2xquery [32,33]. Note that queries in each batch may not be the same since they are for different datasets.

5.5.2. Evaluation Results and Analysis

In the results of the experiments, the ground truths of most of the requests are empty, which indicates that no matching results existed for these requests. This was expected, since device descriptions and queries were generated randomly and independently. What we were most interested in was how OWL inference affects the quality of the query results. We expected to see differences in the matching results of the two approaches. The queries that did not return any matches were not used in the comparisons of the two approaches. This section shows the quality metrics and analysis. The raw data for the metrics results are available online (https://github.com/YankeeChen/evaluator/tree/master/evaluationresults, accessed on 15 November 2022).

Figure 5a–c show the average recall, precision, and F-Measure metrics of the two approaches for each of the four batches of queries. For each batch, six results are shown. The five bars show, from left to right, the associated metrics for the OWL-based approach with the progressively less complete test ontologies ($B_1$ to $B_5$, from left to right). The rightmost bar shows the associated metric obtained using the XML-based approach.

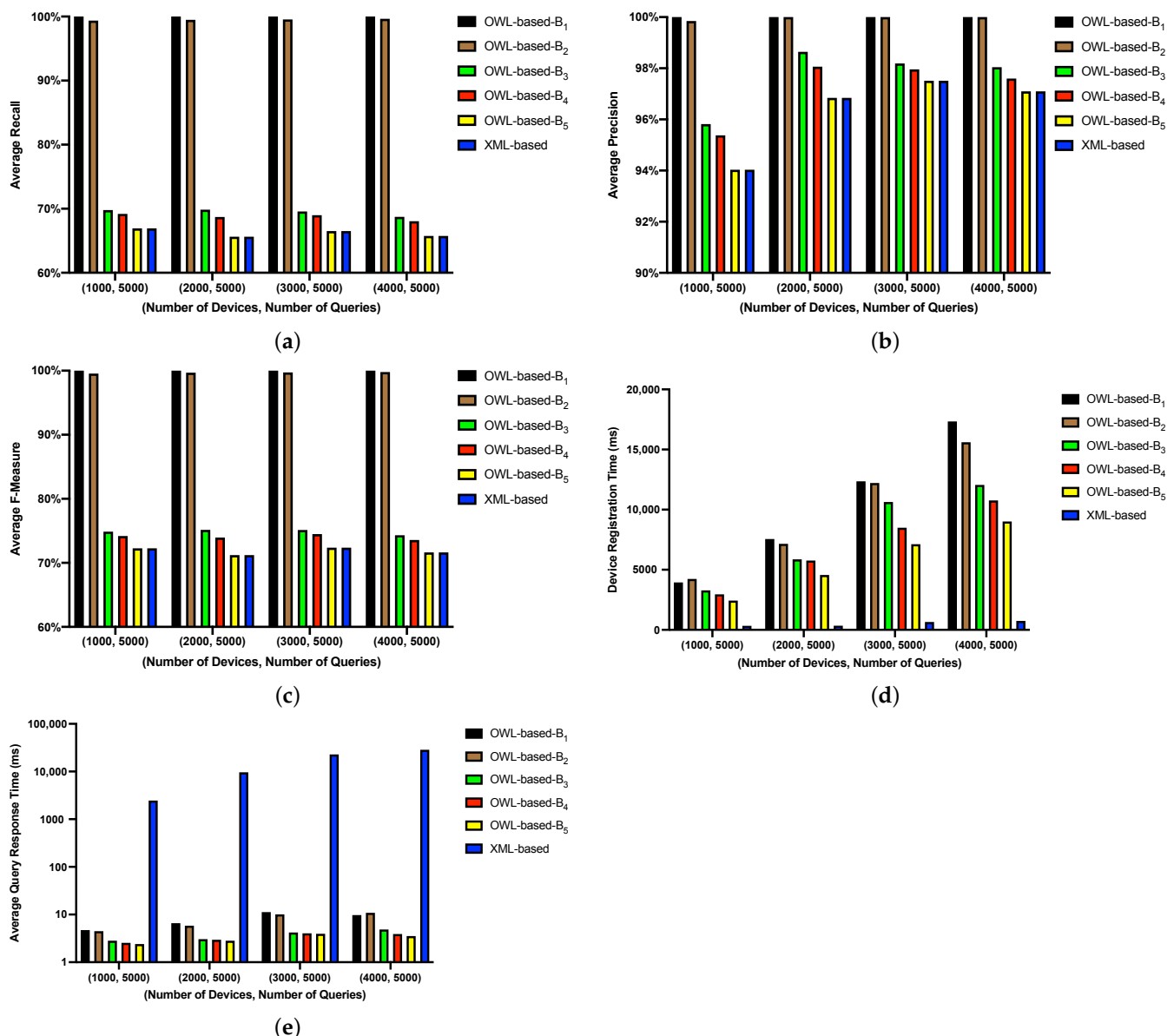

**Figure 5.** Metrics evaluation results (**a**) Average recall comparison; (**b**) Average precision comparison; (**c**) Average F-Measure comparison; (**d**) Device registration time comparison; (**e**) Average query response time comparison.

For the recall metric, we conclude that the XML-based approach provides matching results that are approximately 34% less complete than those produced with the OWL-based approach with the relatively most complete test ontology $B_1$. The result is roughly the same regardless of the number of devices. For instance, the average recall metric value of 5000 queries against 4000 devices in the XML-based approach is 65.75%, which is 34.25% lower than that produced with $B_1$, and the results are about the same for other batches. It can also be observed from the results that the XML-based approach derives exactly the same results as the OWL-based approach with the least complete ontology $B_5$ for each batch. This is as expected, since $B_5$ does not include any OWL axioms except for vocabularies (see in Table 4), and thus no new facts can be derived by the OWL reasoner. Therefore, it derives exactly the same matching devices as the XML-based approach. When it comes to the comparison of the OWL-based approach with different versions of test ontologies, as expected, a more specific ontology (more precise definitions of classes) provides more power (more axioms) to the OWL reasoner, which results in more complete matching.

This is essentially in agreement with the statement that James Hendler once made *"A little semantics goes a long way"* [50].

Regarding the precision metric, as expected, the XML-based approach derives exactly the same precision results as the OWL-based approach with the least complete ontology $B_5$, regardless of the number of devices. The precision metric results of all batches are very high (more than 93%). Similarly to the recall metric results, a richer ontology provides better precision results. It can also be observed that the precision metric results increase with the number of devices until there are more than 3000. This is also as expected, since a query is more likely to have matching results when the datasets are large and diversified enough.

As a combined metric of precision and recall, the F-measure metric results reflect the overall metric results of precision and recall. Thus, similar conclusions and observations apply.

Figure 5d shows the time taken to register devices in the two approaches. As can be seen from the plots, the device registration time in the two approaches increases linearly with the number of devices. For each batch, the device registration time in the XML-based approach is always shorter than that in the OWL-based approach. For the OWL-based approach, apart from registering device descriptions, it takes additional time to run the inference engine and insert inferred facts into the triple store. The richness of the test ontologies affects the registration time; richer ontologies require the OWL reasoning to spend more time on the device registration process.

The plot in Figure 5e shows the average time taken to answer a query by the matchers of the two approaches. We conclude that the query response time in the XML-based approach is exponentially dependent on the number of devices and is significantly longer than that in the OWL-based approach. However, we cannot conclude that XQuery is less efficient than SPARQL. Apart from the query language, the query response time also depends on other factors, such as the query processor, query optimization, dataset structure, etc. Research on these topics is beyond the scope of this paper. Regarding the OWL-based approach, the query response time increases linearly with the number of devices. For each batch in the plot, the query processing time increases with the richness of the test ontologies. The experimental results are reasonable, since richer test ontologies may derive more inferred facts, which results in the expansion of the search scope of each query to derive matching results.

## 6. Discussion

The objective of this section is to discuss how the method satisfies the requirements stated in Section 1.2 and why the techniques used in the method are a good fit for our problem.

Clearly, the main challenge is to assure that the proposed method is fair to the two representation formalisms—XML and OWL. The first step is to use the same inputs and the same evaluation metrics. The problem is that the inputs and the queries must be expressed in two different languages. To assure a relatively good level of fairness, we had to rely on mappings between the languages that were as good as possible. We relied on the fact that OWL uses XML as one of the syntaxes. Thus, the same randomly generated device descriptions were expressed in the XML syntax of OWL and used by both XML and OWL based matchers. Second, queries were expressed in SPARQL and then translated using the xql2xquery tool, whose translation completeness and correctness is assured [32,33]. The same metrics (recall, precision and F-measure) were used to assess both approaches.

One of the most difficult issues was the ground truth—how to assess whether a given selection of the devices to satisfy a given query is correct. In our approach we used a "relative" rather than "absolute" definition of ground truth. Since our objective was to assess the value added by OWL inference, we used the most complete knowledge base as the reference and then compared the performance of the OWL-based matcher using progressively less complete knowledge bases. This approach did not impact the fairness of the comparison between the matchers, since the XML-based matcher provided the most complete information for each of the cases.

The coverage of the space of device descriptions used in our experiments was evaluated in [8], where it was shown that the SDR ontology used in the experiments provides good coverage of the knowledge of various types of RF devices and has extensive coverage of the OWL axioms. Second, the queries used in our experiments were assessed by various evaluation metrics in [9], and it was shown that the coverage of the query space is good.

Since RODG, SQG, xql2xquery, the OWL-based query processor (DeVISor), and the XML-based query processor (BaseX) can all handle large amounts of data, the method is able to deal with large-sized datasets and query sets and can extend to updates of the background knowledge, datasets, query sets, and metrics.

## 7. Conclusions

This paper proposes a method to compare the OWL-based and XML-based approaches to represent and query cognitive radio capabilities using quantitative metrics. In order to prove the feasibility and the correctness of the method, we developed a proof-of-concept system for the method. Two types of metrics, matching quality metrics and performance metrics, were collected by the system with progressively less complete background knowledge representations and different sized sets of devices and queries. The evaluation results clearly demonstrate the advantages of the OWL-based approach in terms of the quality of matching. The results also demonstrate the benefits of using a more specific ontology to improve the quality of the matching results at the cost of sacrificing some level of performance. To be specific, the quality metrics evaluation results show that (i) the quality of matching in the OWL-based approach is always no worse than the XML-based approach, regardless of the richness of the selected ontology and sizes of the devices, and (ii) in the OWL-based approach, a more specific ontology results in better matching results. (iii) A derived consequence of the above conclusions is that shallow use of OWL does not buy much in terms of the quality of matching.

The performance metrics evaluation results show that (i) the registration process in the XML-based approach takes less time, regardless of the number of devices; (ii) in the OWL-based approach, a more specific ontology results in a longer device registration time; (iii) the query processing time in the XML-based approach increases exponentially with the number of devices and is significantly longer than the query time in the OWL-based approach. However, we cannot decisively conclude that the XQuery query is less efficient than the SPARQL query, since the query processing time also depends on other factors, such as the query processor, query optimization, and dataset structure; and (iv) in the OWL-based approach, the query processing time depends on the richness of the ontology and thus it takes more time to process the same queries against the same devices with richer ontologies. Although we have not performed any investigations of the use of the proposed method in other domains, we suggest that the method is also applicable to areas other than just RF devices.

Following the main conclusion of this paper, the continuation of this research should focus on the OWL-based approach, which provides "more semantics" than XML. This can be achieved in two ways—by using richer ontologies and extensions to OWL using rules. The standardization of ontologies for the communications domain began with the development of the Cognitive Radio Ontology (CRO) by the Wireless Innovation Forum [51]. The CRO was then extended to CRO2 [52](available at [53]). Work on the standardization and extension of the expressive power of OWL is continuing at the IEEE [54].

**Author Contributions:** Conceptualization, Y.C., M.M.K. and J.M.; methodology, Y.C., M.M.K. and J.M.; software, Y.C. and J.M.; validation, Y.C. and M.M.K.; formal analysis, Y.C. and M.M.K.; investigation, Y.C. and M.M.K.; resources, Y.C. and M.M.K.; data curation, Y.C. and M.M.K.; writing—original draft preparation, Y.C. and M.M.K.; writing—review and editing, J.M., K.R.C.; visualization, Y.C. and M.M.K.; supervision, M.M.K.; project administration, Y.C. and M.M.K.; funding acquisition, M.M.K. All authors have read and agreed to the published version of the manuscript.

**Funding:** This research was funded by the Defense Advanced Research Projects Agency (Grant Number W911NF-14-C-0065).

**Institutional Review Board Statement:** Not applicable.

**Informed Consent Statement:** Not applicable.

**Data Availability Statement:** Data are contained within the article. The data presented in this study can be requested from the authors.

**Conflicts of Interest:** The authors declare no conflict of interest.

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
