# Peer review of "Metrics-Based Comparison of OWL and XML for Representing and Querying Cognitive Radio Capabilities"

_applsci, doi:10.3390/app122311946_

Round 1

Reviewer 1 Report

The authors presented a comparison work on wireless devices spectrum access mechanisms using OWL and XML language. They studied the effectiveness of both languages and the OWL-based approach is proven to be more effective than the XML-based approach. The authors should consider the following comments:

1. Please include OWL and XML in full form in the abstract.

2. The motivation for using OWL and XML is not clear. More justifications are needed for selecting these two approaches than other languages.

3. XML is already defined in line 65, not needed in line 127.

4. Some description is necessary for Listing 1 and Listing 2.

5. A table would be added to list the “Experimental Parameters” for reproducing the results.

6. The figures (a), (b), (c) started from 60%,90%,60% and (d) and (e) started from 0. It would be better if all start from 0.

7. The images in Figure 5 are small, please enlarge the image size, it's hard to see the labels in the images.

8. The figures' captions are too small, please include more description in the figure caption.

9. Is there any reference to Eq 12 and 13? How β affect the system?

10. If possible put the related work section at the beginning of the article (after Introduction).

Reviewer 2 Report

Today, many of us use wireless connectivity devices. The number of such devices necessitates effective spectrum allocation, which must be used as efficiently as possible. To do this, it is necessary to properly manage, monitor and allocate resources to users. Appropriate coordination of the activities of devices in the communication network must be maintained. In this case, applications that allow for appropriate management are helpful, the effects of which each user of wireless communication devices can experience on a daily basis.

In order to achieve the assumed interoperability of devices in the network, it is necessary to use a communicator (language) describing radio capabilities, requests, scenarios, policies and spectrum availability.

The authors of the article present a thorough comparison of the use of two languages: OWL and XML. For this purpose, they proposed a method to evaluate the basic parameters: precision, recall, device registration and query response time. The authors carried out a preliminary analysis mentioned languages and gave reasons for choosing these languages.

Despite of my overall positive assessment of the paper I have several remarks:

1. Conclusions include a description of obtained results. But in my opinion this is not sufficient. The paper raises a problem which applies to wide scientific area of wireless communication. I hope these analyses are not finished. Therefore authors have to write several sentences about future works.

2. The paper has to be reorganized. Chapter 6 "Related works" has to move at the beginning of the paper. Then the paper would be better understood.

3. Figure 5: description (a), (b), (c), (d) and (e) should be moved to main caption of the figure

4. Section 3.2: Metrics Formulation or Metrics Formalization?

5. Indexes k, n, m are used in equations. Meaning of the indexes one should explain.

Round 2

Reviewer 1 Report

I have no more comments.